# Unveiling Source of Performance Variance on Search-based Compiler Optimization

Pranati Modumudi, Xiyou Zhou, and Sunghyun Park, OctoML

*Abstract*—To squeeze out the performance in the post-Moore era, the compiler auto-tuning approach has been widely studied and productized. Despite its superior efficiency in compiler optimization problems, performance variance in final tuning output has long been an issue for search-based auto-tuning methods. It poses a challenge to research reproducibility and production stability. In general, the causes of such performance variance come from many aspects across different system layers. In addition to generic causes, we observe that auto-tuners add unique sources of variance, including the use of different search methods and cost models.

In this work, we specifically focus on the performance variance originating from the nature of auto-tuning. Based on our observation, we set three major hypotheses on the search method, cost model, and hardware characteristics. Then, we validated our hypotheses through experiments with a production auto-tuner and a representative set of machine learning workloads. Our preliminary result suggests impactful factors to consider in future investigations.

## I. Introduction

As the performance from hardware innovations is stagnating due to the end of Moore's Law [10], leveraging latent opportunities in compiler optimization space is becoming more critical. However, the ever-increasing complexity in hardware and software stacks makes even the most advanced compilers fail to deliver the best optimization settings for individual workloads from time to time. It strongly motivates the recent efforts in auto-tuning approach in both industry and academia [1], [6], [8], [9], [11], [12], [14]–[16].

Most auto-tuning techniques are essentially an iterative feedback-directed search in the optimization space - promising candidates are generated, evaluated, and reflected upon to generate even more promising candidates for future tuning trials. Evaluation is usually conducted by the actual compilation and run-time measurement for the sake of high accuracy. However, due to its expensive cost, recent studies introduce various cost models that can select good candidates without the actual measurement [2], [5], [13]. One popular approach is to train a learning-based cost model on-the-fly by using the run-time samples collected during the tuning process. And then, auto-tuning methods can use this cost model to filter out discouraging candidates cheaply and evaluate only the promising ones with the actual measurement. In addition, to effectively explore the extremely huge and complicated optimization search space, these auto-tuning methods adopt intelligent search methods to make the best use of the feedback from the evaluation. Evolutionary search [16], multi-armed bandit [14], ensemble search [1] and reinforcement learning [6], [9] are representative examples.

Overall, the auto-tuning approach has successfully demonstrated its strong performance and is widely deployed in the industry [6], [11], [12], [15], [16]. However, performance variance in the final tuning outcome has been problematic and one of the major challenges to reproducibility and production stability of auto-tuning methods. The source of variance can be diverse and laid across the multiple system layers ranging from hardware to software stacks (e.g., non-deterministic behavior of hardware, interrupt from the operating system, etc.). Also, auto-tuning approach adds another source of variance. For example, since most search methods are randomization-based, the search pattern may differ across tuning runs.

To address this problem, we investigate the representative root causes for the performance variance. Specifically, this work focuses on the unique sources of variance that originated from the nature of the auto-tuning approach. Based on our experience with production auto-tuners, we set three hypotheses on the search method, cost model, and hardware property. Then, we conduct experiments to verify each of the hypotheses. Given the importance of the problem, we believe our preliminary result is a meaningful step forward to attack a long-standing concern in auto-tuners.

The contributions of this paper are as follows:

- We discuss the unique source of performance variance in the popular auto-tuning approach and set three hypotheses to assess their influence.
- We analyze our hypotheses with the production auto-tuner and the representative machine learning workloads on different hardware devices.
- Future works outline our next steps to address this enormous challenge in the auto-tuning approach.

## II. Potential Sources of Performance Variance in Compiler Auto-tuners

Figure 1 illustrates the overarching workflow of the popular auto-tuning approach with a learning-based cost model [2], [5], [16]. Once users provide the tuning budget (e.g., number of candidate trials), auto-tuners repeatedly perform four steps using its three major components: search method, cost model, and measurer. First off, a search method, such as evolutionary search [7], generates a set of promising candidates and queries the cost model to predict the competency between candidates and filter out non-promising ones. Then, the measurer takes the filtered candidates and evaluates them on the actual hardware. It accompanies the compilation that applies the optimization setting in each candidate and the run-time measurement of the resulting executable. Once the run-time performance is

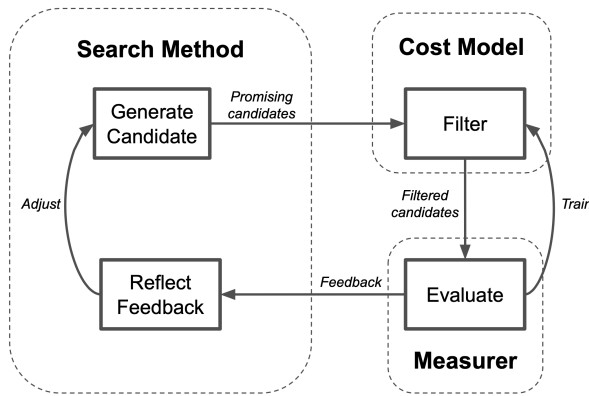

Fig. 1: Representative auto-tuning workflow with three major components: search method, cost model and measurer. Within the predefined tuning budget (e.g., number of candidate trials, wall-clock time), these components in the auto-tuner repeat the four steps by working together closely.

collected, it will be provided as feedback to both the search method and cost model. The search method will use the run-time performance as the feedback to generate more promising candidates in the next iteration (e.g., mutation and cross-over in the evolutionary search [7]). On the other hand, the cost model will use the run-time performance for its training process.

Because every component interacts very closely with each other, a slight difference in the behavior of one component may lead to quite different outcomes across tuning runs. For example, popular search methods [1], [6], [7] leverage random search to a certain extent for their statistical efficiency, especially at the beginning of the tuning process. Also, unfortunately, some source of non-determinism is inevitable - it is impossible to completely eliminate run-time noise during the measurement. Therefore, in this work, we focus on three aspects that we may be able to improve in the future auto-tuner design.

*A. Hypothesis 1: search method may generate imbalanced training data for the cost model.*

We observe the potential dilemma between the search method and the cost model. In order to find out the best candidates, the search method exploits the previous feedback to effectively cut down the search space and focus on the narrowed promising space. Ironically, this may imply the generation of biased training data for the cost model.

*B. Hypothesis 2: instability of cost model accuracy may affect final performance variance.*

Since the cost model is constructed online, it is not clear if training data collected during the tuning process is enough to make the cost model's accuracy mature and stable enough. Also, the cost model would populate the different training data across the tuning runs due to the randomness in the search method. *Hypothesis 1* adds another source of instability in its accuracy.

*C. Hypothesis 3: certain hardware and its target-specific optimization may inherently have higher run-time variance than others.*

Non-determinism in hardware behavior incurs some run-time noise in the measurer and results in noisy feedback that may guide the search in a biased direction. If certain hardware is intrinsically noisier, the auto-tuner may consider this factor in its strategy of utilizing feedback. Also, different hardware targets use different sets of optimizations and some of the optimizations (e.g., memory optimization) might be more susceptible to such variance.

## III. Evaluation

*A. Experiment Setup*

*1) Auto-tuner:* For experiments, we investigate the new auto-tuning technology based on TVM [4] called MetaSchedule. While following the auto-tuning workflow in Figure 1, MetaSchedule is designed to provide a convenient tuning experience with both template-based search [5] and template-free search [1].

MetaSchedule takes a set of customizable schedule rules where each schedule rule defines the decision space of compiler optimization(s). Before the search kicks off, its space generator will use those rules to express the valid search space for each workload. By default, MetaSchedule provides schedule rules for CPU and GPU targets that apply important compiler optimizations, such as vectorization, multi-level loop tiling, loop unrolling and etc. Please note that MetaSchedule operates in the later compilation pipeline so the tensor layout, operator fusion, and lowering decisions are already handled from the earlier stages.

To effectively traverse the tremendous search space, MetaSchedule adopts the search strategy that identifies its promising subspace and generates good candidates. For example, the evolutionary search strategy [7] examines the results of candidates from the previous trials to choose the performant settings and utilizes them to generate even more promising ones by applying techniques like a crossover. A random search strategy is also a popular option as it explores every direction of the search space in a uniform way without leveraging any feedback during the tuning process.

In order to maximize auto-tuning efficiency, MetaSchedule constructs to disregard the unpromising candidates by predicting their performance XGBoost [3] model based on the extracted workload features.

*2) Workloads:* We select five representative tensor programs to conduct experiments, covering most compute-intensive subgraphs in end-to-end model tuning. The workloads are as follows:

- *C2D:* 2-D Convolution, NHWC layout
- *C3D:* 3-D Convolution, NHWC layout
- *GMM:* Batch Matrix Multiplication
- *T2D:* Transposed 2-D Convolution, NHWC layout
- *TBG:* Transposed Batch Matrix Multiplication

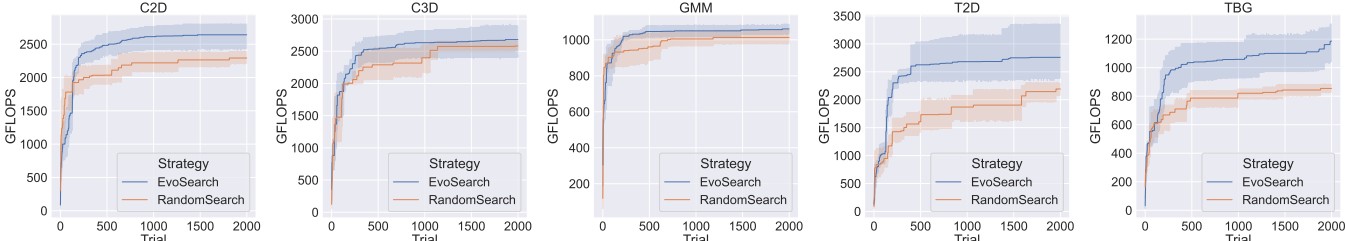

(a) Examine *Hypothesis 1* on CPU: Comparison of evolutionary search (default search method in MetaSchedule, denoted in *EvoSearch*) and random search. Random search shows more stable results across tuning runs despite its lower performance.

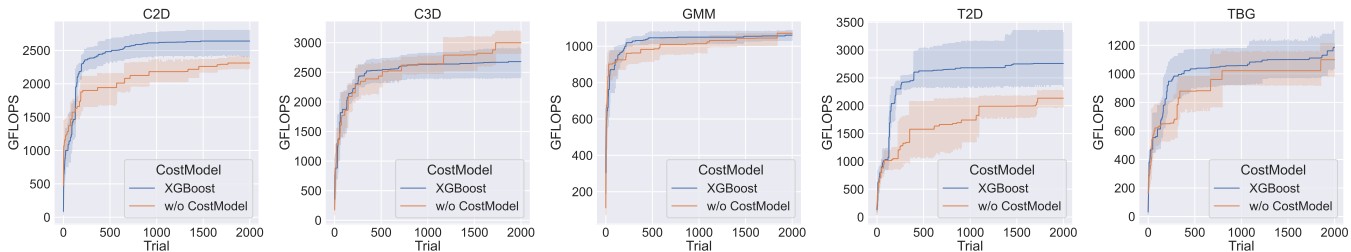

(b) Examine *Hypothesis 2* on CPU: Comparison of evolutionary search with XGBoost [3] (default cost model in MetaSchedule) and evolutionary search without cost model (no filtering step in Figure 1). Variance seems lower when the cost model is disabled. Interestingly, final performance is pretty close on *GMM* and *TBG*. On *C3D*, the search without any cost model even outperforms the one with the cost model.

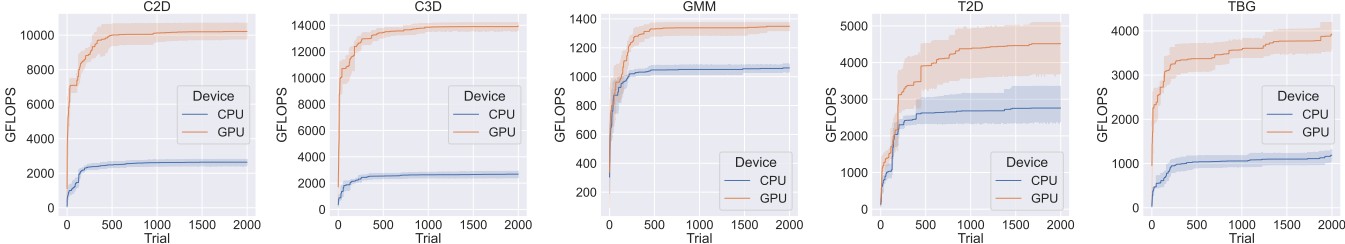

(c) Examine *Hypothesis 3*: Comparison of the variance between CPU and GPU experiments. In every workload, the CPU showed more stable performance despite its lower performance.

Fig. 2: Our preliminary experiments to validate our hypotheses. We plotted their tuning progress by targeting five representative deep learning workloads and collecting the best performance in GFLOPS at each trial in the tuning budget. AWS instances of `c5.9xlarge` (36 Intel Xeon Platinum 8223CL vCPUs) and `p3.2xlarge` (NVIDIA V100 GPU) are used for CPU and GPU experiments correspondingly. The shaded area shows the performance variance (standard variation) at each trial.

*3) Hardware:* For experiments, we used AWS EC2 cloud machines. CPU experiments were conducted on `c5.9xlarge` machine with 36 Intel(R) Platinum 8223CL CPU @ 3.00GHz vCPUs. GPU experiments were conducted on `p3.2xlarge` machine with an NVIDIA V100 GPU.

For the stable measurement, we separate autotuning from measurement - a single AWS Cloud machine (`c5.12xlarge`) is assigned to proceed with the main tuning process and trigger measurement on the remote slave machines via TVM's RPC module. This approach also parallelizes the repeated measurement across the slave machines and speeds up the overall tuning time significantly.

### B. Experimental Result

Figure 2 presents our preliminary results. With a tuning budget of 2,000 trials of candidate evaluation (with measurer),

we stamped the performance of the best candidate identified up until each trial and plotted them to show the tuning progress. As the performance metric, GFLOPS is chosen given its standardness. To quantify the performance variance, we launched five independent tuning jobs and measure their standard deviation at trial. Overall, we observed that tuning jobs experience high variance during early phases and stabilize their performance as the tuning proceeds. However, in some tuning workloads, such as *T2D*, the variance gets larger. Our future study will examine this further by looking into its kernel implementation and assigning more trials.

- *Hypothesis 1: search method may generate imbalanced training data for the cost model.*

  To validate this hypothesis, we compare the tuning

progress between evolutionary search (*EvoSearch*) and random search. Since random search explores search space without any feedback, it would generate the most balanced training data that can be collected during the tuning process. Figure 2a presents the experimental result. Since random search does not exploit any feedback from previous trials, it generally shows worse performance than evolutionary search. However, it consistently demonstrates its stability by exhibiting less performance variance across workloads. Especially, random search showcases significantly better stability in *C3D*, *T2D* and *TBG*. Future work will experiment different ratios of feedback exploitation and random search.

- *Hypothesis 2: instability of cost model accuracy may affect final performance variance.*

  For comparison, we conduct tuning runs with an XG-Boost cost model and without it (pure measurement). Without any cost model, all promising candidates generated by evolutionary search will be evaluated with measurer (See Figure 1). Note that these two experiments had an equal number of measurements during each tuning instance (i.e., 1 trial = 1 measurement), so the overall tuning time stays similar as overhead in the measurer usually dominates the overall tuning time. Figure 2b presents the result.
  Overall, we could stabilize the performance variance by disabling the cost model. With the cost model, the variance on most workloads (i.e., All but *GMM*) does not improve and gets worse often. We suspect this might be attributed to the unstable cost model accuracy with the online training approach. Future work will try pre-trained cost model to see its impact.

- *Hypothesis 3: certain hardware and its target-specific optimization may inherently have higher run-time variance than others.*

  Figure 2c exhibits default MetaSchedule tuning (i.e., evolutionary search with cost model) on two separate hardware architectures – CPU (Intel Xeon Platinum 8223CL) and GPU (NVIDIA V100) machines. Although GPU shows higher performance throughput in every workload, it generally presents worse variance. Especially, variance in *T2D* is significant.
  Such difference in hardware performance variance could be attributed to the difference during search space construction - some schedule rules for compiler optimization could be hardware-specific, especially layout-related rules. Additionally, we suspect that the different memory architectures in CPU and GPU may result in distinct performance sensitivity on tensor program compilation. Minor changes in GPU memory access patterns may bring an enormous performance difference compared to CPU. Future work will set up the identical optimization search space to different hardware and further explore

their characteristics.

## IV. Future Direction and Conclusion

This work investigates the performance variance problem in the production compiler auto-tuning approach. By focusing on the causes of such variance that originates from the nature of auto-tuning, we suggest three potential factors to consider to alleviate the performance variance in the future design of auto-tuners: Firstly, the search method may need to balance the statistical efficiency of the search process and the quality of training data to construct the cost model. Future studies will examine the ratio of randomly generated candidates and see its impact on the balance of training data and try to develop a better search method that can provide high performance with less variance. Secondly, we may need to take a deeper look at the training progress of the cost model. If a standard tuning budget is insufficient to reach stable accuracy, we may consider offline tuning or transfer learning. Last but not least, we show that certain hardware architecture and its target-specific compiler optimizations can be inherently noisier than others. Our future investigation will design more sophisticated experiment settings to observe the impact of each optimization and the behavior of underlying hardware components.

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
