# OpenReview forum: "Unveiling Source of Performance Variance on Search-based Compiler Optimization"
_iscaconf.org/ISCA/2022/Workshop/MLArchSys — MLArchSys 2022_

### Official Review · Reviewer_vius · 2022-05-10
**Interesting and important topic, but paper is too narrow in scope**

**Rating:** 5
**Confidence:** 5

**Review:**

This paper discusses the sources of variance in cost model-driven autotuning. This is a great topic, and I am pleased to see work in this area. However, I feel that this paper is not yet quite ready for publication. In particular, I feel that the current suite of experiments is too shallow for a reader to derive insight that can be used to inform best practices.

The paper describes the sources of variance in terms of three hypotheses. For each hypothesis, a single experiment is run comprising two choices, over a limited set of workloads (convolution and matrix multiplication with differing dimensionalities, and with transposition on or off), and results plotted over five runs. I think a much larger set of choices for each of those variables is required:

- There is no quantifications for variances observed except for eyeballing the size of the error bars in Fig 2. Since the paper frames the sources in terms of hypotheses, I would like to see hypothesis tests and more rigorous statistical analysis.
- Since results vary greatly from workload to workload, I would like to see a broader set of workloads used.
- For 2(a), how much randomness is there in EvoSearch? How does varying degrees of exploration/exploitation affect outcomes?
- For 2(b), how does varying the degree of online/offline training data for the cost model affect outcomes?
- For 2(c), the two hardware architectures are so distinct that it is hard to draw any conclusions from the data. I would like a much more fine-grained analysis. Does variance change from one minor architecture revision to another? Does changing the number of cores and available memory affect CPU results?
- Hypothesis 3 states that optimization choices may affect variance. How is this evaluated?
- It is unclear to me whether MetaSchedule is SoA, or whether findings on this tuning approach would transfer to other works. Could the experiments be applied to any of the cited autotuning works? IMO replicating the experimental setup of a prior work would more impactful and convincing.

I would love to see the authors expand the scope and depth of their experiments, and would encourage them to submit a revised version to another venue. A thorough and methodical examination of the effects of variance on autotuning results would be an impactful work.

---

### Official Review · Reviewer_EXpU · 2022-05-19
**I find the insights generated from this short paper interesting and I think the paper will serve as a good topic of conversation.**

**Rating:** 8
**Confidence:** 3

**Review:**

# Paper Summary:
This paper contributes an experimental evaluation that captures the extent of performance variance introduced in the process of auto-tuning compiler optimizations.  The authors give a background description of how current compiler auto-tuning solutions operate and identify three potential sources of performance variance introduced by the different components of such auto-tuners. Their analysis shows that parameters such as the exploration strategy of the search space, the use of an online cost model, and the choice of the underlying hardware technology, can significantly impact the variance of the performance levels enabled by a compiler auto-tuning solution.

# Review Summary:
I find the insights generated from this short paper interesting and I think the paper will serve as a good topic of conversation. The results presented raise questions that can be relevant to the overall workshop audience and across related problems, such as the general impact on performance variance of the underlying hardware technology and the use of randomness in software pipelines. More specifically, here is what I enjoyed in this paper and some suggestions for improvements:

# Things I enjoyed:
* The introduction nicely describes the workflow of compiler auto-tuners and gives the necessary background information for the non-expert reader.
* The paper is nicely structured into sections and the graphs are easy to comprehend.
* I find the experimental results interesting, especially because the workloads and parameters explored show that there can be cases with significant variance in performance. In particular, the results with CPU vs. GPU execution or the case of not using a cost model are interesting and can be applicable to many systems problems, not just compiler auto-tuning.

# Suggestions for improvements in the results presented:
* If possible, sharing more information on the workloads regarding their code structure or data access patterns will help the authors reveal more insights as to why we see a different behavior for certain workloads, such as T2D over GPU.
* Consider changing the y-axis of your experiments to better capture the impact on performance introduced by the variance, so that you can argue that one parameter may introduce x% or x times performance variance and show how that is not trivial for the target workloads.
* Since you have 3 concrete different hypothesis, I would have liked to see 3 concrete answers that summarize the major insight from each experiment.

# Suggestions for improvements in the document structure:
* It will help to make an additional grammar pass on the document, there are sentences that can be improved, there are a lot of missing “the” etc.
* There is no need to repeat the experimental methodology in the caption of Figure 2.
* Some suggested edits to some titles:

II. Potential sources of performance variance in compiler auto-tuners.

IIIA. Experimental Setup.

IIIB. Experimental Results.

---

### Official Review · Reviewer_Sq9g · 2022-05-23
**This paper investigates three potential source of high variance in compiler auto-tuning.**

**Rating:** 5
**Confidence:** 4

**Review:**

This paper suggests three potential sources for high variance in the outcome of compiler auto-tuning, which is empirically evaluated on different tensor programs and one auto-tuning method. The first and second sources of high variance are very similar and are attributed to the performance of the cost model which is due to lack of balanced training data or sufficient training. The third source of variance is the inherent noise in hardware measurements. Overall, the observations are valid, however, they are somewhat intuitive/trivial: a good auto-tuning system is expected to have a "good" cost model. Additionally, the explanations and the trends in the plots are not very well-correlated, i.e., the behavior explained in the text is not necessarily observed very strongly in the provided plots. Finally, while the first and second source of variance may be possible to address to a degree, it is unclear what can be done regarding the third one, i.e., hardware characteristics.

Editorial comments: Please explain the x and y axes in your plots very clearly in the text, i.e., what are trial and GFLOPs? How does GFLOPs correlate with the performance of the autotuner?

---

### Decision · Program_Chairs · 2022-05-30

**Decision:**

Accept

**Comment:**

The reviewers overall found this line of research important and interesting, but suggested the authors to provide more detailed evaluations. In particular, expanding the benchmark suite to a broader set of diverse workloads with distinct characteristics is critical to the claims in the paper. Furthermore, the paper would be much better if the authors provide additional ablation studies that clarify the limitations of this work.

Because of the overall positive feedback from the reviewers, the paper is an `Accept` for presentation at the workshop.